# Perioperative outcomes and hospitalization costs of radical vs. conservative surgery for hepatic cystic echinococcosis: A retrospective study

Honggang Shi[1], Kahaer Tuerxun[2], Aizizaimu Yusupu[2], Zainuer Yusupu[3], Qilin Xu[2], Yibin Jia[4], Musitaba Maimaitireyimu[2], Tuerhongaji Maimaitiaili[2], Muzaipaer Muhetajiang[2], Jiaxin Lin[5], Chengmin Ma[1], Xiaofeng Li[1]*, Yuanquan Wu[2]*, Yonghui Su[6]*

1 Department of Gastroenterology, The Fifth Affiliated Hospital, Sun Yat-sen University, Zhuhai, Guangdong, China, 2 Department of Hepatobiliary Surgery, The First People's Hospital of Kashi Prefecture, Kashi, Xinjiang, China, 3 Department of Ultrasound Medicine, The First People's Hospital of Kashi Prefecture, Kashi, Xinjiang, China, 4 Department of General Surgery II, People's Hospital of Jiashi County, Jiashi, Kashi, Xinjiang, China, 5 Department of Infectious Diseases, The Fifth Affiliated Hospital, Sun Yat-sen University, Zhuhai, Guangdong, China, 6 Department of Gastrointestinal Surgery, The Fifth Affiliated Hospital, Sun Yat-sen University, Zhuhai, Guangdong, China

☯ These authors contributed equally to this work.
* lixiaofeng@mail.sysu.edu.cn (XL); doctor_wuyq@163.com (YW); suyh@mail.sysu.edu.cn (YS)

**Data Availability Statement:** All relevant data are within the manuscript and its Supporting Information files.

## Abstract

### Background

Surgical intervention is a crucial treatment for hepatic cystic echinococcosis. However, the choice between radical and conservative surgery remains controversial. This study aimed to compare the perioperative outcomes and hospitalization costs between radical and conservative surgery for hepatic cystic echinococcosis.

### Method

A retrospective cohort study was conducted on patients undergoing surgical treatment at the First People's Hospital of Kashi Prefecture from July 1, 2012, to October 1, 2023. Propensity score-matching analysis was utilized to mitigate patient selection bias between the two surgical groups.

### Result

Of the 434 patients included, 324 underwent conservative surgery and 110 underwent radical surgery. After propensity score-matching, 182 conservative surgery and 102 radical surgery patients were compared. Radical surgery patients experienced longer operative time, higher intraoperative blood loss, increased blood transfusion, and higher costs compared to conservative surgery patients. However, no differences were observed in short-term outcomes, including overall morbidity, death, bile leak, effusion, pulmonary infection, incision infection, intestinal obstruction, ICU stay, abdominal drainage time, and postoperative hospital stays.

**Funding:** This work was supported by the Youth Science and Technology Top Talent Project - Cultivation of Grassroots Science and Technology Core Talents of the Xinjiang Uygur Autonomous Region (Grant No. 2022TSYCJC0007) (to K.T.), the Project of Comprehensive Plan for Talent Assistance to Kashi Prefecture of Xinjiang, 'The Zhu Jiang Scholars, Tianshan Talents' Collaborative Expert Studio Innovation Team Plan of First People's Hospital of Kashi (Grant No. KDYY-202203) (to K.T.), and the Science and Technology Program of Kashi Prefecture (Grant No. KS2022020) (to Y.J.). The funders had no role in study design, data collection and analysis, decision to publish, or preparation of the manuscript.

**Competing interests:** The authors have declared that no competing interests exist.

## Conclusion

This study suggests that radical surgery is associated with greater surgical complexity and higher hospitalization costs, while it doesn't offer a significant short-term advantage. Conservative surgery may be a viable option in resource-limited settings or for patients unsuitable for complex procedures. Further research with long-term follow-up is needed to determine the optimal approach.

## Author summary

Surgical outcomes and hospitalization costs are crucial considerations for surgeons when selecting a treatment approach. This retrospective study compared perioperative outcomes and hospitalization costs between radical and conservative surgery for hepatic cystic echinococcosis. We analyzed data from patients treated at the First People's Hospital of Kashi Prefecture from July 1, 2012, to October 1, 2023, using propensity score matching to minimize selection bias.

Our findings revealed that while radical surgery entailed longer operative times, greater blood loss, and higher costs, there were no significant differences in short-term outcomes such as mortality, complication rates, or length of hospital stay compared to conservative surgery. This suggests that conservative surgery may be a viable alternative, especially in resource-limited settings where the complexity and high costs of radical surgery pose challenges.

Importantly, our study focused on short-term outcomes and hospitalization costs. Future research should incorporate long-term follow-up to assess the recurrence rates associated with each surgical approach.

## Introduction

Cystic echinococcosis (CE) is a zoonotic infectious disease caused by *Echinococcus granulosus sensu lato*, with the liver being the most commonly affected organ [1,2]. Large cysts may lead to serious complications, such as cystobiliary communication and acute cholangitis [3,4]. Ruptures of CE cysts can induce symptoms such as urticaria, fever, and even anaphylactic shock [5], and may also lead to the dissemination of the parasite. The global annual expenditure on CE is estimated to exceed 760 million US dollars [6], exacerbating the economic burden in endemic regions.

Management of hepatic cystic echinococcosis (HCE) is complex and includes surgical treatments, percutaneous treatment, chemotherapy, and "watch and wait" [2,5,7]. Surgical treatments are categorized mainly into radical and conservative approaches. Radical surgery (RS), such as hepatectomy and total cystectomy, offers the advantage of completely removing the lesion with a low risk of recurrence [8–12], but it is more complex and traumatic compared to conservative surgery (CS), like sub-total cystectomy and partial cystectomy [13–15].

Currently, significant controversy still exists regarding the intraoperative and short-term postoperative outcomes for RS and CS[16–19], and there is also limited research on the associated costs. Many studies are limited by factors such as insufficient adjustment for confounding biases.

Kashi, located in the western part of China, is an endemic area for HCE and has considerable experience in the standardized diagnosis and treatment of the disease [20,21]. This study

aims to provide a real-world comparison of the perioperative outcomes and hospitalization costs associated with radical and conservative surgery to the management of HCE. We hypothesize that RS, compared to CS, will result in higher risks and costs without significant differences in short-term perioperative outcomes based on our experience.

## Methods

### Ethics statement

The study adhered to the ethical principles of the Declaration of Helsinki's, received approval from the Ethics Committee of the First People's Hospital of Kashi Prefecture (Ethics Number: [2024] Fast Review Research (27)), and did not require patient consent forms, as the data were anonymized.

### Clinical data analysis

A retrospective analysis was conducted on the clinical data of patients who underwent surgical treatment for HCE in the Department of Hepatobiliary and Pancreatic Surgery, the First People's Hospital of Kashi Prefecture from July 1, 2012, to October 1, 2023. Inclusion criteria were patients diagnosed with HCE based on clinical history, physical examination, and imaging (Color Doppler Ultrasound or CT), who subsequently underwent surgical treatment. Patients with incomplete medical records (specifically missing preoperative Color Doppler Ultrasound results, surgical records, or postoperative clinical course records) were excluded. The data were uniformly collected after the patients were discharged, and only the data from the first surgery were included for each patient in the study.

We collected clinical data from patients' hospital records from the medical record system. The patients were divided into the RS group and the CS group based on the surgical method performed. Demographic and clinical characteristics, such as age, gender, cyst location, number of cysts, cyst diameter, WHO cyst classification [22,23], epigastric pain, abdominal mass, history of abdominal surgery, fever, extrahepatic cyst, and open abdominal surgery, were collected for both groups. We extracted cost data for each patient from the hospital's admission records and categorized total cost into the following groups: surgery-related costs, non-surgery-related costs, consumables costs, medication costs, examination costs, and other costs (Total). Detailed definitions of the indicators can be found in S1 Text. The number of cysts was divided into two groups: equal to 1 and greater than 1, because single and multiple cysts have different impacts on study outcomes. Building on clinical expertise and prior research findings [14,17,24], we hypothesize that these indicators may influence the recovery process and the occurrence of postoperative complications in surgical patients. Therefore, we selected preoperative and intraoperative data as confounding factors during the propensity score matching (PSM). The preoperative data included age, gender, cyst location, cyst diameter, number of cysts, WHO cyst classification, epigastric pain, abdominal mass, fever, and extrahepatic cysts. The intraoperative factor was open abdominal surgery. The outcomes of our study were divided into two categories: intraoperative and postoperative. The intraoperative indicators comprised operative time, blood loss, and blood transfusion, all of which were documented in the surgical records. Postoperative indicators included overall morbidity, death, bile leak, effusion, pulmonary infection, incision infection, intestinal obstruction, ICU stay, abdominal drainage time and postoperative hospital stay, which were sourced from medical records and imaging studies. Abdominal drainage time was divided into two groups: less than or equal to 7 days and greater than 7 days. This categorization was based on clinical observations and represents different rates of postoperative recovery. Detailed definitions of the indicators can be found in S1 Text.

The hepatobiliary surgeons at our hospital determined the surgical approach through discussions based on the condition of the patient's cyst (such as its location, proximity to major vessels, and size), in conjunction with local medical standards. These decisions were also agreed upon by the patient and their family. All patients received general anesthesia, and the majority underwent laparotomy. In the CS group, the primary surgical steps included protecting surrounding tissues, managing cyst contents, and resecting most of the cyst layers. The RS group's procedures primarily involved removing all the layers of CE cyst or resecting part of liver tissues. Following surgery, patients in the CS group routinely received albendazole at a dosage of 10–15 mg/kg/day for three months. Postoperative abdominal drainage tubes were placed in all patients.

## Statistical analysis

Baseline characteristics of continuous variables were reported as the mean ± standard deviation (SD) if they followed a normal distribution, and as the median (interquartile range, IQR) otherwise. Categorical variables were presented as frequencies (%). In our analysis, three individuals were excluded due to extensive missing data, which precluded meaningful analysis. No missing data issues were observed in the remaining subjects.

To mitigate the impact of potential confounders, we utilized PSM to equalize covariates across treatment groups. This ensures that differences observed in outcomes are more likely attributed to the treatments rather than confounding factors. Propensity scores were calculated using logistic regression, incorporating relevant preoperative demographic and clinical characteristics, as well as the intraoperative factor of open abdominal surgery. We prioritized matching patients from the RS group, which had fewer patients. We performed 1:2 greedy nearest neighbor matching with a caliper of 0.1, using the R package MatchIt. The "greedy" approach means matches were selected one by one without optimizing a broader criterion, thus focusing on immediate pair suitability. After performing PSM, we assessed the differences in baseline characteristics between the excluded and retained patients to evaluate the impact of the matching process. We specifically compared the cyst diameters and other relevant covariates using appropriate statistical tests. For normally distributed continuous variables, we applied independent t-tests, while for non-normally distributed variables, we used Mann-Whitney U tests. Categorical variables were analyzed using chi-square or Fisher's exact tests. The results showed that there were differences in cyst diameter and location between the excluded and retained patients ($P < 0.05$), while there were no differences in age, gender, and WHO classification (P = 0.11, 0.99, 0.21)

Following matching, we compared clinical outcomes between the two groups using appropriate statistical tests. For normally distributed continuous variables, we used analysis of variance (ANOVA); for non-normally distributed variables, we used non-parametric tests. Categorical variables were assessed using the chi-square test or Fisher's exact test. Sensitivity analyses were performed by varying the caliper width and matching algorithm. Further details are provided in S1 Table. We defined statistical significance as a p-value less than 0.05. Additionally, to evaluate the adequacy of our sample size, a post-hoc power analysis was conducted for each measure. The power analysis was performed before and after PSM using R software (version 4.2.2) with an alpha level of 0.1. The other statistical analyses in this study were conducted using R software, version 4.2.2, with the MatchIt package, and MSTATA software (www.mstata.com).

## Result

A total of 434 patients were included in this study, with 324 undergoing CS and 110 undergoing RS. After PSM, 182 patients in the CS group and 102 in the RS group were analyzed, as

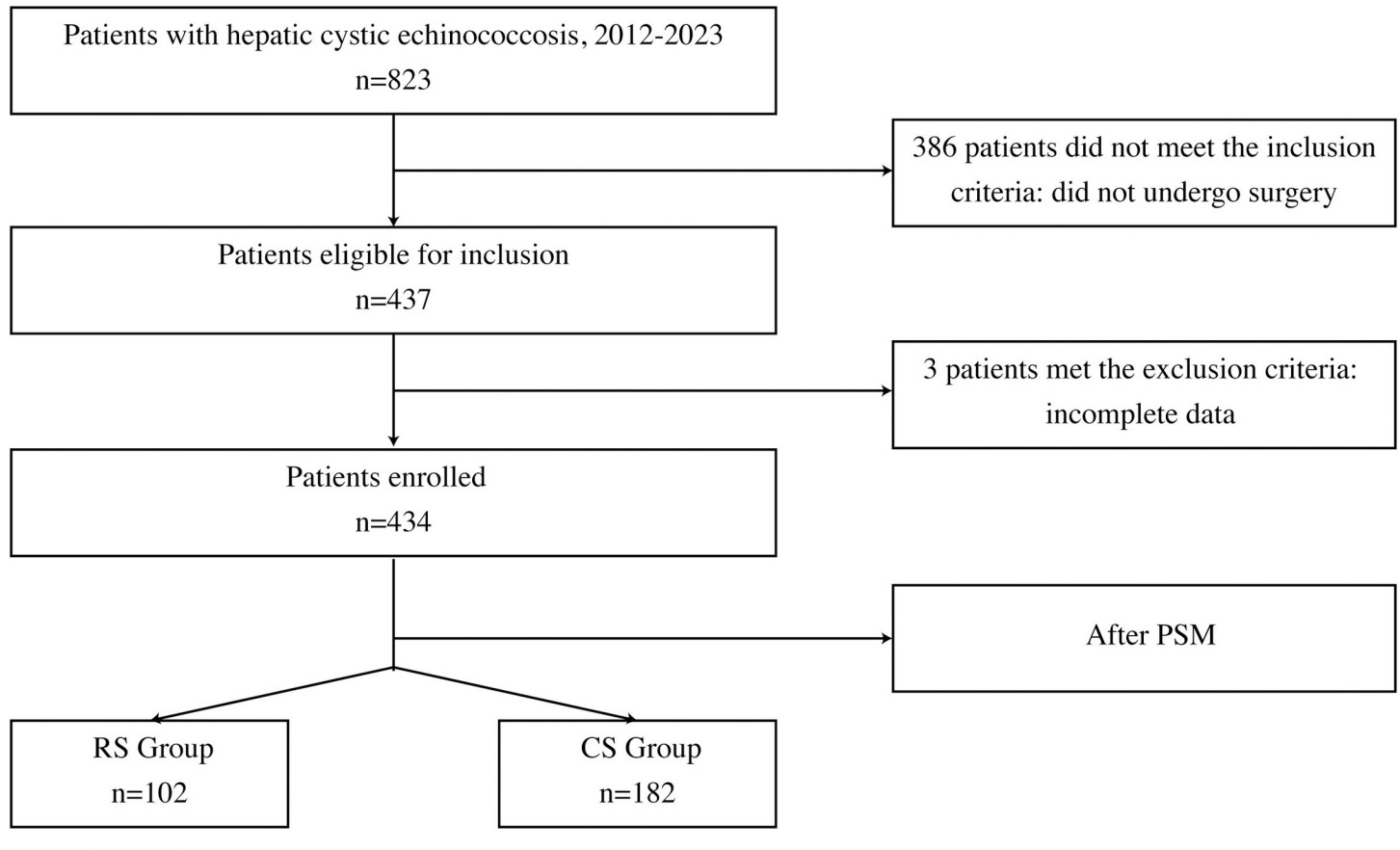

**Fig 1. Patient Recruitment Process.**

shown in Fig 1. This study evaluated the statistical power of different outcomes before and after PSM:operative time (0.99–0.99), blood loss (0.44–0.20), blood transfusion (0.06–0.05), overall morbidity (0.11–0.11), bile leakage (0.58–0.28), effusion (0.29–0.18), pulmonary infection (0.16–0.07), death (0.27–0.08), ICU stay (0.14–0.25), abdominal drainage time($>$7) (0.86–0.55), postoperative hospital stay(0.47–0.22), incision infection(0–0), surgery cost (0.99–0.98), total cost 0.99–0.90).

Table 1 summarizes the baseline characteristics of both groups, showing the covariate balance before and after PSM. After matching, the standardized mean differences (SMDs) significantly decreased and were all less than 0.1, indicating effective balancing of covariates between groups. This enhanced balance facilitated a more reliable comparison of treatment effectiveness by minimizing the potential influence of confounding factors.

Table 2 displays intraoperative indicators, such as operation time, blood loss, and blood transfusions. After matching, significant differences in these parameters were observed between the CS and RS groups.

As illustrated in Table 3, the postoperative outcomes of the two surgical strategies were compared in the entire cohort and within the matched populations. No significant differences were observed in additional outcomes such as overall morbidity, bile leak, effusion, pulmonary infection, incision infection, intestinal obstruction, abdominal drainage time, and length of hospital stay post-operation. Notably, the abdominal drainage time was statistically significant before matching (P = 0.007) but lost significance after matching (P = 0.076). The total costs for the RS group were higher than those for the CS group. The differences are primarily attributed

**Table 1. Baseline covariates before and after matching.**

| Variables | Level | Before Matching | | | After Matching | | |
|---|---|---|---|---|---|---|---|
| | | CS | RS | SMD△ | CS | RS | SMD△ |
| n | | 324 | 110 | | 182 | 102 | |
| Age (mean (SD)) | | 36.22 (17.77) | 35.63 (19.60) | -0.030 | 35.27 (18.02) | 34.90 (19.67) | 0.015 |
| Gender (%) | Male | 136 (42.0) | 52 (47.3) | 0.106 | 76 (41.8) | 47 (46.1) | 0.088 |
| | Female | 188 (58.0) | 58 (52.7) | -0.106 | 106 (58.2) | 55 (53.9) | -0.088 |
| Cyst location (%) | Left lobe | 58 (17.9) | 32 (29.1) | 0.246 | 47 (25.8) | 27 (26.5) | -0.022 |
| | Right lobe | 234 (72.2) | 65 (59.1) | -0.267 | 110 (60.4) | 62 (60.8) | 0.050 |
| | Both the lobes | 32 (9.9) | 13 (11.8) | 0.060 | 25 (13.7) | 13 (12.7) | -0.046 |
| Cyst diameter (cm, mean (SD)) | | 10.36 (3.65) | 9.30 (3.57) | -0.297 | 9.52 (3.09) | 9.38 (3.64) | -0.039 |
| Number of cysts (%) | 1 | 231 (71.3) | 78 (70.9) | -0.009 | 124 (68.1) | 72 (70.6) | 0.054 |
| | >1 | 93 (28.7) | 32 (29.1) | 0.009 | 58 (31.9) | 30 (29.4) | -0.054 |
| WHO cyst classification (%) | CE1 | 155 (47.8) | 46 (41.8) | -0.122 | 77 (42.3) | 43 (42.2) | -0.010 |
| | CE2 | 50 (15.4) | 25 (22.7) | 0.174 | 37 (20.3) | 24 (23.5) | 0.023 |
| | CE3a/b | 54 (16.7) | 19 (17.3) | 0.016 | 33 (18.1) | 17 (16.7) | -0.026 |
| | CE4 | 65 (20.1) | 18 (16.4) | -0.100 | 35 (19.2) | 18 (17.6) | 0.013 |
| | CE5 | 0 (0.0) | 2 (1.8) | 0.136 | 0 (0.0) | 0 (0.0) | 0.000 |
| Epigastric pain (%) | Yes | 183 (56.5) | 64 (58.2) | 0.034 | 104 (57.1) | 60 (58.8) | 0.010 |
| | No | 141 (43.5) | 46 (41.8) | -0.034 | 78 (42.9) | 42 (41.2) | -0.010 |
| Abdominal mass (%) | Yes | 166 (51.2) | 62 (56.4) | 0.103 | 103 (56.6) | 58 (56.9) | -0.020 |
| | No | 158 (48.8) | 48 (43.6) | -0.103 | 79 (43.4) | 44 (43.1) | 0.020 |
| History of abdominal surgery (%) | Yes | 59 (18.2) | 25 (22.7) | 0.108 | 40 (22.0) | 21 (20.6) | -0.023 |
| | No | 265 (81.8) | 85 (77.3) | -0.108 | 142 (78.0) | 81 (79.4) | 0.023 |
| Fever (%) | Yes | 3 (0.9) | 1 (0.9) | -0.002 | 2 (1.1) | 1 (1.0) | 0.000 |
| | No | 321 (99.1) | 109 (99.1) | 0.002 | 180 (98.9) | 101 (99.0) | 0.000 |
| Extrahepatic cyst (%) | Yes | 29 (9.0) | 10 (9.1) | 0.005 | 17 (9.3) | 8 (7.8) | -0.034 |
| | No | 295 (91.0) | 100 (90.9) | -0.005 | 165 (90.7) | 94 (92.2) | 0.034 |
| Open abdominal surgery (%) | Yes | 317 (97.8) | 101 (91.8) | -0.220 | 176 (96.7) | 99 (97.1) | 0.036 |
| | No | 7 (2.2) | 9 (8.2) | 0.220 | 6 (3.3) | 3 (2.9) | -0.036 |

△Standardized Mean Difference

to surgery-related costs (P<0.001), consumables costs (P = 0.003), and examination costs (P = 0.042). For further details, please refer to S2 Table.

Sensitivity analyses were performed by varying the caliper width and matching algorithm, which confirmed the robustness of the main findings. Further details are provided in S1 Table.

**Table 2. Intraoperative indicators before and after matching.**

| | Before Matching | | | After Matching | | | |
|---|---|---|---|---|---|---|---|
| | CS | RS | P | CS | RS | Mean/media Difference (95%CI) | P |
| n | 324 | 110 | | 182 | 102 | | |
| Operative time (min, Mean (SD)) | 135.8 (48.77) | 177.2 (84.47) | <0.001 | 135.9 (51.24) | 177.3 (85.50) | 41.44 (25.54, 57.35) | <0.001 |
| Blood loss (ml, Median (IQR)) | 50 (50, 150) | 100 (50,200) | <0.001 | 50 (50,100) | 100 (50,200) | 50 (25,50) | <0.001 |
| Blood transfusion (ml, Median (IQR)) | 0 (0,0) | 0 (0,0) | <0.001 | 0 (0,0) | 0 (0,0) | 0 (0,0) | 0.002 |

**Table 3. Immediate perioperative outcomes before and after matching.**

| | Before Matching | | | After Matching | | | | |
|---|---|---|---|---|---|---|---|---|
| | CS | RS | P | CS | RS | Risk Ratio (95% CI) | Odds Ratio (95% CI) | P |
| n | 324 | 110 | | 182 | 102 | | | |
| Overall morbidity (%) | 100 (31%) | 35 (32%) | 0.85 | 56 (31%) | 33 (32%) | 1.05 (0.74–1.5) | 1.08 (0.64–1.81) | 0.78 |
| Bile leak (%) | 36 (11%) | 6 (5.5%) | 0.083 | 17 (9.3%) | 6 (5.9%) | 0.63 (0.26–1.55) | 0.61 (0.23–1.59) | 0.31 |
| Effusion (%) | 65 (20%) | 27 (25%) | 0.32 | 38 (21%) | 25 (25%) | 1.17 (0.75–1.83) | 1.23 (0.69–2.19) | 0.48 |
| Pulmonary infection (%) | 16 (4.9%) | 4 (3.6%) | 0.57 | 7 (3.8%) | 3 (2.9%) | 0.76 (0.20–2.89) | 0.76 (0.19–3.00) | >0.99 |
| Death (%) | 0 (0%) | 1 (0.9%) | 0.25 | 0 (0%) | 1 (0.9%) | - | - | 0.36 |
| ICU stay (%) | 8 (2.5%) | 4 (3.6%) | 0.51 | 3 (1.6%) | 4 (3.9%) | 2.38 (0.54–10.42) | 2.44 (0.53–11.10) | 0.26 |
| Abdominal drainage time > 7 days (%) | 201 (62%) | 52 (47%) | 0.007 | 102 (56%) | 46 (45%) | 0.80 (0.63–1.03) | 0.64 (0.40–1.05) | 0.076 |
| Postoperative hospital stays (day, mean (SD)) | 9.4 (3.21) | 8.9 (2.50) | 0.119 | 9.2 (3.13) | 8.9 (2.55) | -0.28 (-1.00, 0.43) * | | 0.434 |
| Incision infection (%) | 1 (0.3%) | 0 (0%) | >0.99 | 1 (0.5%) | 0 (0%) | - | - | >0.99 |
| Intestinal obstruction (%) | 1 (0.3%) | 0 (0%) | >0.99 | 0 (0%) | 0 (0%) | - | - | - |
| Total cost (USD, mean (SD)) | 3013.57 (6425.59) | 3402.23 (8940.21) | <0.001 | 2992.07 (6960.82) | 3382.13 (8840.85) | 2775.35 (911.78, 4638.93) * | | 0.004 |

△U.S. Dollar; * Mean Difference (95%CI)

## Discussion

This retrospective study compared the perioperative outcomes and hospitalization costs between CS and RS for HCE. Our cohort included 324 patients who underwent CS and 110 patients who underwent RS. We found that RS is associated with greater surgical complexity, including longer operative times and increased blood loss, along with higher costs. Also, there are no differences in overall morbidity, bile leak, and other complications.

CS and RS are two surgical strategies for treating HCE. CS involves incomplete cystectomy, offering the advantages of less wound trauma and lower surgical difficulty, while preserving more healthy liver tissue. In contrast, RS aims to completely remove the lesion to reduce the risk of disease recurrence. However, due to the larger scope of lesion removal in RS, doctors in resource-limited areas may face noteworthy challenges such as higher surgical difficulty and increased hospitalization costs. Our study aims to provide further insight by comparing the short-term perioperative outcomes and hospitalization costs of CS and RS among patients in Western China. Meanwhile, we used the statistical method of PSM to reduce selection bias and improve the comparability between the two groups.

Our study first compared the intraoperative conditions between the two groups. The results showed that RS had a longer operation time than CS. We attribute this to the extensive dissection in RS, which affects numerous blood vessels and increases complexity. This is consistent with previous research [8,9,11,18,25,26]. Concurrently, we observed that the RS group had more intraoperative bleeding than the CS group, which was expected. This finding is supported by several studies [17,26] including Waad Farhat's [8] which reported a higher incidence of intraoperative bleeding in the RS group through PCA analysis. While some studies have indicated that blood loss can impact short-term prognosis following hepatectomy [27], the median blood loss in our study for both groups was less than 250ml. This may be the reason why the increased blood loss in the RS group did not affect the short-term prognosis.

Our study examined the postoperative complications, which are a key clinical indicator of our concern. After applying PSM, the overall morbidity rate was 32.4% in the RS group and

30.2% in the CS group, with no statistical difference between them (P = 0.71). This finding is consistent with the results reported by H. O. El Malki [18] and Sami Akbulut [25], who also concluded the same after PSM and multivariable logistic analysis of 493 patients. In contrast, Waad Farhat [8] observed a significant reduction in overall morbidity in the RS group following a paired comparative analysis of 914 patients. We speculate that the discrepancies in the findings may stem from differences in the study populations and methodologies. Waad Farhat's research [8] included patients treated at Sahel Hospital from January 2000 to December 2019, whereas the current study involved patients from Western China between July 1, 2012, and October 1, 2023. Additionally, the current study used PSM, while Waad Farhat employed a paired comparative analysis. Bile leak is a common postoperative complication. The overall bile leak rate in the study of 434 patients was 9.7%. After applying PSM, the bile leak rate in the CS group was 9.3%, which nearly matches the 10.1% reported in a previous review [14,24]. Despite a lower bile leak rate in the RS group compared to the CS group after PSM, the difference was not statistically significant (P = 0.31). This indicates that RS did not reduce the risk of postoperative bile leaks in this study. This finding aligns with recent results from Huang [17] in Linzhi, China, and is comparable to those observed in laparoscopic surgery [9]. Numerous studies indicate that RS may reduce the incidence of postoperative bile leaks [8,11], as it involves biliary reconstruction in healthy tissue during hepatectomy, which may lower the bile leak rate [14]. However, our study did not observe this expected reduction in bile leaks with RS. Interestingly, some studies have identified an enlarged hepatic resection surface as a significant factor in postoperative bile leaks [28]. Additionally, increased blood loss during hepatectomy may cause bile duct ischemia, potentially leading surgeons to overlook sites prone to bile leaks [29]. Moreover, RS encompasses total cystectomy, which is distinct from hepatectomy, and could lead to varying bile leak rates. Therefore, it cannot be conclusively assumed that RS reduces the incidence of bile leaks. Future studies should aim to further specific types of RS and their impact on bile leak.

The study additionally investigated other short-term postoperative prognostic indicators, including mortality, effusion, pulmonary infection, incision infection, intestinal obstruction, ICU stay, drainage time > 7 days, and postoperative hospital stays. We observed no significant differences in these outcomes between RS and CS groups. Before PSM, there was a notable difference in drainage time > 7 days between the RS and CS groups. However, this difference dissipated after bias adjustment through PSM. Among the enrolled patients, one case resulted in death the day after surgery. Using Fisher's exact test, we found no significant difference in mortality rates between the two groups. The patient's death was caused by *Echinococcus granulosus sensu lato* invading the inferior vena cava, which led to fatal bleeding in the ICU. Considering that the patient's hospital and drainage days totaled only one day which contradicted the poor prognosis, we excluded this patient from our analysis and reanalyzed the data. The results remained consistent, reaffirming the robustness of our findings.

We also explored hospitalization costs associated with the two surgical approaches. We found RS group faced higher surgical and total costs compared to the CS group. Based on our cost analysis, the higher total cost of RS is mainly due to the significantly higher surgery-related costs (P < 0.001), consumables Costs (P = 0.003), and examination costs (P = 0.042). It is important to note that in the study setting, national health insurance policies cover a substantial portion of the costs for the surgical treatment of HCE. However, in the absence of such comprehensive health insurance policies, patients may be more inclined to choose the less costly CS.

Recurrence is a critical indicator of the efficacy of surgical interventions for HCE. One major limitation of this study is the lack of long-term follow-up data which could affect the interpretation of our findings. While this study primarily focuses on short-term outcomes,

existing literature offers important insights into recurrence rates. Several studies have indicated that radical surgery is generally associated with lower recurrence rates [8,11]. However, the study by Jaén-Torrejimeno et al. suggests that the type of surgery (whether total or partial cystectomy) is not significantly related to an increased risk of relapse [30]. Furthermore, a propensity score-matched comparison of laparoscopic surgeries also revealed that, although the recurrence rate was lower for radical surgeries, the difference was not statistically significant. The recurrence rate for endocystectomy has been reported to be 4.8%, which supports the potential effectiveness of conservative treatment [14]. Future studies should include extended follow-up periods to comprehensively evaluate the long-term outcomes of different surgical approaches and to determine the most appropriate surgical method based on individual patient characteristics. Ultimately, the selection of a surgical approach for an individual patient should consider multiple factors including cyst size, quantity, location, patient health condition, quality of life expectations, and recurrence concerns. We emphasize that no single surgical method can address all cases, and each technique has its specific advantages, disadvantages, and indications. Our study offers evidence comparing the short-term outcomes and hospitalization costs of the two surgical methods, which can help physicians make informed clinical decisions.

This study has certain clinical limitations. Firstly, this is a retrospective study, and biases mainly stem from selection bias and information bias. Selection bias arises from the decisions made by doctors to choose different treatment options for different patients, while information bias stems from potential errors in medical record documentation. Although we used PSM to reduce selection bias, it cannot eliminate them entirely. There were significant differences in cyst diameter and location between the excluded and non-excluded patients ($P < 0.05$), which may limit the generalizability of the study results to patients with larger cysts. Secondly, the low statistical power of some indicators is due to the low incidence and small differences of some complications in clinical practice. Future research should consider increasing the sample size to enhance statistical power, ensuring the reliability and generalizability of the results.

## Conclusion

Our study demonstrates that RS is associated with increased surgical complexity and higher hospitalization costs compared to CS. However, we found no significant difference in short-term outcomes between the two approaches. This suggests that CS may be a viable alternative, particularly in resource-limited settings or for patients who are not suitable for more complex procedures. Further randomized controlled trials with long-term follow-up are needed to determine the optimal surgical approach for HCE.

## Supporting information

**S1 Data. Primary data.**
(XLSX)

**S1 Text. The definitions of indicators.**
(DOCX)

**S2 Text. Checklist.**
(DOC)

**S1 Table. Sensitivity analysis.**
(DOCX)

**S2 Table. Cost comparison between RS and CS.**
(DOCX)

**S3 Table. Baseline characteristics of patients with P.**
(DOCX)

## Acknowledgments

The authors express their sincere gratitude to the medical and nursing staff of the Department of Hepatobiliary Surgery for their significant contributions to the patients and this study. We are deeply thankful to the patients who provided the data necessary for this research. Additionally, we appreciate the valuable guidance provided by the hospital's Ethics Committee on the research protocol.

## Author Contributions

**Conceptualization:** Honggang Shi, Kahaer Tuerxun, Xiaofeng Li, Yuanquan Wu, Yonghui Su.

**Data curation:** Honggang Shi, Aizizaimu Yusupu.

**Formal analysis:** Aizizaimu Yusupu, Zainuer Yusupu.

**Investigation:** Honggang Shi, Aizizaimu Yusupu, Zainuer Yusupu, Qilin Xu, Yibin Jia, Musitaba Maimaitireyimu, Tuerhongaji Maimaitiaili, Muzaipaer Muhetajiang, Jiaxin Lin.

**Methodology:** Kahaer Tuerxun, Yonghui Su.

**Project administration:** Kahaer Tuerxun, Xiaofeng Li, Yuanquan Wu, Yonghui Su.

**Resources:** Yibin Jia, Musitaba Maimaitireyimu, Tuerhongaji Maimaitiaili, Muzaipaer Muhetajiang, Jiaxin Lin.

**Software:** Honggang Shi, Aizizaimu Yusupu, Chengmin Ma.

**Supervision:** Kahaer Tuerxun, Xiaofeng Li, Yuanquan Wu, Yonghui Su.

**Validation:** Tuerhongaji Maimaitiaili, Muzaipaer Muhetajiang, Jiaxin Lin, Chengmin Ma.

**Visualization:** Zainuer Yusupu, Qilin Xu, Yibin Jia, Musitaba Maimaitireyimu.

**Writing – original draft:** Honggang Shi.

**Writing – review & editing:** Kahaer Tuerxun, Xiaofeng Li, Yuanquan Wu, Yonghui Su.

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
