## [Decision Letter · Decision Letter 0]

18 Jul 2024

Dear Su,

Thank you very much for submitting your manuscript "Comparative analysis of perioperative outcomes and costs between radical versus conservative surgical interventions for hepatic cystic echinococcosis: A retrospective propensity score-matching cohort study" for consideration at PLOS Neglected Tropical Diseases. As with all papers reviewed by the journal, your manuscript was reviewed by members of the editorial board and by several independent reviewers. In light of the reviews (below this email), we would like to invite the resubmission of a significantly-revised version that takes into account the reviewers' comments. 

We cannot make any decision about publication until we have seen the revised manuscript and your response to the reviewers' comments. Your revised manuscript is also likely to be sent to reviewers for further evaluation.

Sincerely,

Francesca Tamarozzi

Section Editor

Francesca Tamarozzi

Section Editor

Editor's revision

MAJOR

Methods: 

1) The cost assessment part is basically lacking. The authors must write the detailed methods for the cost estimation. 

2) Data must be clearly presented regarding timing of data collection in relation to surgery. If more than one surgery was carried out during the period, it must be specified how data were handled. Info in S2 is insufficient. In S2 are also not specified important factors for surgical decision, such as location of the cyst (e.g. adjacent to a big vessel, superficial vs deep-seated cyst), which I would expect influence both the choice of surgery and possible complications. Please explain in the text why these factors were not considered or, if considered, how were handled.

3) line 133: “overall morbidity”. In S2 the definition is quite broad...please be more specific about complications included ad what "adverse outcomes" includes

Results: cyst classification (e.g. Table 1) The WHO-IWGE classification does not include “Type 1, Type 2, Type 3, Type 4 and Type 5” but CE1, CE2, CE3a, CE3b, CE4, CE4. Consider also that “CE3” per se does not exists, but there are CE3a (with detached parasitic layers) and CE3b (partially solid with daughter cysts). Please apply throughout the correct cyst classification.

Discussion: 

1) in line 217 the authors state that there were no significant differences in short-term clinical outcomes between the groups, but later on (line 226), on the contrary, they state that radical surgery had “higher risks”. Please clarify

2) The limitations of the study should be more clearly indicated and highlighted. For example, in line 232: the authors mention “beyond recurrence rate”: this is not a secondary thing but a very strong limitation of this study because recurrence is a serious issue and often surgeons who do not follow the patients for a long time are completely unaware about the recurrences and think all their work was perfect, therefore this severe study limitation must be stressed as such. Also, despite the matching process, there are 150 excluded and this could have biased the results. 

Please provide in the methods any possible way the EXCLUDED people may have been systematically excluded and/or different from the included patients and in the discussion stress this big number of excluded patients and its consequences also a major limitation of the study

3) Line 308-309 and 326-327: the sentences are completely unclear: why CS would be better than RS “considering the absence of data about recurrences”? What if all CS induce recurrence? Just with the data presented this cannot be so strongly concluded. Considering that this seems the core of the message from this study, this needs to be thoroughly clarified.

4) Line 323-324: the authors seem implying that there are patients for which the risk of recurrence is warranted. This seems awkward to say the least. Please clarify what you mean

OTHER:

Abstract and line 67 and line 91-92 : surgery is not the primary treatment for hepatic cystic echinococcosis. The WHO recommends a stage-specific approach for uncomplicated hepatic CE cysts, that takes also into consideration their size, and therefore cysts are recommended to be treated with albendazole alone (for example small CE1 or CE3a), percutaneous procedures (CE1, CE3a), or not treated at all and just followed-up over time (CE4-CE5). Please amend to convey a correct information

Line 93-96: please use the current terminology for surgery (ref doi: 10.1051/parasite/2020024.) here and throughout the text

Line 84: E granulosus in italic

Line 87: dissemination is not a symptom

Line 113: what is “color ultrasound”?

Line 114: what exact list of incomplete data would make the patient excluded from the analysis? 

Line 144: what does “internal cysts” means?

Line 219: the authors state that RS would provide more “comprehensive solution” for HCE. However, the authors did not investigate relapses, therefore they cannot drive this consideration from their study. If the drive it from other studies, they must be referenced. Please amend the sentence and clarify

Line 222: what characteristics would have the patients “with a low risk of recurrence”?

Line 338-340: please rephrase for clarity and explain the beaning of the biases you mention and in what direction such biases might have influenced your results

Reviewer's Responses to Questions

**Key Review Criteria Required for Acceptance?**

**Methods**

-Are the objectives of the study clearly articulated with a clear testable hypothesis stated?

-Is the study design appropriate to address the stated objectives?

-Is the population clearly described and appropriate for the hypothesis being tested?

-Is the sample size sufficient to ensure adequate power to address the hypothesis being tested?

-Were correct statistical analysis used to support conclusions?

-Are there concerns about ethical or regulatory requirements being met?

Reviewer #1: Everything is OK.

Reviewer #2: methods are sound and correct:

The study retrospectively analyzed 434 patients who underwent surgical treatment for HCE at the First People's Hospital of Kashi Prefecture between July 1, 2012, and October 1, 2023. Patients were divided into two groups based on the surgical method: radical surgery (RS) and conservative surgery (CS). PSM was employed to adjust for potential confounding factors, ensuring balanced covariates between the groups. Statistical analyses, including ANOVA, chi-square tests, and Fisher's exact test, were conducted to compare intraoperative and postoperative outcomes, as well as associated costs.

Reviewer #3: (No Response)

**Results**

-Does the analysis presented match the analysis plan?

-Are the results clearly and completely presented?

-Are the figures (Tables, Images) of sufficient quality for clarity?

Reviewer #1: Yes

Reviewer #2: results are clear and reasonably well presented.

The study revealed several key findings:

Intraoperative Outcomes: RS was associated with longer operative times, higher intraoperative blood loss, and increased need for blood transfusion compared to CS.

Postoperative Outcomes: No significant differences were found between RS and CS in terms of overall morbidity, mortality, bile leakage, effusion, pulmonary infection, incision infection, intestinal obstruction, ICU stay, abdominal drainage time, or postoperative hospital stays.

Costs: RS incurred higher surgical and total costs compared to CS.

Reviewer #3: (No Response)

**Conclusions**

-Are the conclusions supported by the data presented?

-Are the limitations of analysis clearly described?

-Do the authors discuss how these data can be helpful to advance our understanding of the topic under study?

-Is public health relevance addressed?

Reviewer #1: Yes

Reviewer #2: the conclusions are fine, including statements about a few limiting flaws of the design

The study concluded that while RS is associated with greater intraoperative risks and higher costs, it does not offer significant short-term clinical advantages over CS. This finding suggests that CS might be a viable alternative, particularly in resource-limited settings where the risks and costs associated with RS are less manageable. The results highlight the importance of individualized treatment plans and the need for further randomized controlled trials with long-term follow-up to guide the optimal surgical approach for HCE.

Strengths:

Robust Methodology: The use of PSM to control for confounding variables strengthens the reliability of the comparative analysis.Comprehensive Data: The study provides a thorough analysis of both intraoperative and postoperative outcomes, as well as cost considerations.

Limitations

Retrospective Design: The study's retrospective nature may introduce biases related to data collection and patient selection.

Short-term Focus: The study focuses primarily on short-term outcomes, with limited information on long-term recurrence rates and overall patient prognosis.

 Sample Size: Some statistical analyses may be underpowered due to the relatively small sample size for certain outcomes.

Reviewer #3: (No Response)

**Editorial and Data Presentation Modifications?**

Reviewer #1: Comments for Line 44: "Surgical intervention is the primary treatment for hepatic cystic echinococcosis" and Line 90: "The primary treatment for hepatic cystic echinococcosis (HCE) involves surgical method".

Please include in the abstract that today for uncomplicated hepatic CE and according to WHO classification and suggestions they are 4 options: medical treatment with albendazol; percutaneous; surgery (open or laparoscopic) and watch and wait. And for complicated CE hepatic cysts surgery is the treatment of choice.

Line 146: Following surgery, patients in the CS group routinely received albendazole at a dosage of 10-15 mg/kg/day for three months.

Comment: do you have the data of preoperative albendazole. If you got, please include it.

In line 299 and 329 and Fig 1, word "hydatid" is used 

Please adjust the nomenclature according to: 

Vuitton DA, McManus DP, Rogan MT, Romig T, Gottstein B, Naidich A, Tuxun T, Wen H, Menezes da Silva A; World Association of Echinococcosis. International consensus on terminology to be used in the field of echinococcoses. Parasite. 2020;27:41. doi: 10.1051/parasite/2020024. Epub 2020 Jun 3. PMID: 32500855; PMCID: PMC7273836.

Table 1: WHO type 1/2/3/4. 

Please use appropriate classification: CE1/CE2/CE3a/CE3b/CE4/CE5.

I noted that they are many cases CE4 cysts under surgical treatment. Why? Usually, CE4 type cysts can be managed with watch and wait strategies. 

Please include in the abstract that in regions with limited resource according to results, CS might be a more appropriate choice.

Reviewer #2: I just suggest to avoid use of Chinese currency to describe the costs of procedures. I would prefer a more international one, just to allow a better understanding

Reviewer #3: (No Response)

**Summary and General Comments**

Reviewer #1: Dear authors. Is a very interesting and important paper. During many years RS was thinking as the ideal treatment by the lower rate of recurrence as main argument. But as you mentioned in limited or resource-constrained environments CS is a very good, best or even unique option for many patients. The resources and surgical skills for RS are without any doubt higher than CS. In RS usually you will need ICU (intensive care unit). So with CS, you can made possible to treat surgically many patients without needed of ICU. 

Also I agree, as you mentioned that prospective study with longterm follow up is needed.

Reviewer #2: The study provides valuable insights into the perioperative outcomes and costs associated with RS and CS for HCE. The findings suggest that CS may be a more suitable option in resource-limited settings, given its comparable short-term outcomes and lower costs. However, the study also underscores the need for further research to assess long-term outcomes and guide clinical decision-making in the management of HCE.

Overall, this paper contributes significantly to the understanding of surgical management for HCE, offering evidence-based guidance for clinicians in choosing the most appropriate surgical approach for their patients.

Reviewer #3: In this retrospective study, the authors examine the differences in perioperative outcomes and costs between radical surgery (RS) and conservative surgery (CS) for the treatment of hepatic cystic echinococcosis. Using data from 434 patients treated at a single center, the study found that while RS is associated with longer operative times, greater blood loss, increased blood transfusions, and higher costs, it does not provide significant short-term clinical advantages over CS. The study concludes that CS may be a viable alternative, particularly in resource-limited settings.

The manuscript is generally well written with clear and concise language. The study addresses an important and ongoing debate in the surgical management of hepatic cystic echinococcosis and provides a thorough comparison of perioperative outcomes and costs between radical and conservative surgical approaches. The use of propensity score matching to mitigate selection bias is a strength and contributes to the robustness of the findings. 

However, there are some important issues and questions which should be addressed: 

1. The title is descriptive but can be shortened for clarity. Consider: "Perioperative Outcomes and Costs of Radical vs. Conservative Surgery for Hepatic Cystic Echinococcosis: A Retrospective Study".

2. How were surgical and total costs calculated? The method of calculation and cost details should be reported in Methods and results. 

3. Please consider P-value for the result of statistical test in Table 1.

4. It is unclear, which factors (preoperative? Intraoperative? Etc.) were used for PSM. Since P-value was not reported in preoperative data (table 1), the rationality of PSM is questionable. A PSM itself can cause selection bias, when not indicated. Maybe it is better to report real-world data. 

I recommend major revision before considering for publication.

PLOS authors have the option to publish the peer review history of their article (what does this mean?). If published, this will include your full peer review and any attached files.

Reviewer #1: Yes: Leonardo Javier Uchiumi

Reviewer #2: Yes: Marcello Maestri MD PhD

Reviewer #3: Yes: Sepehr Abbasi Dezfouli
---

## [Decision Letter · Decision Letter 1]

10 Oct 2024

Dear Su,

We are pleased to inform you that your manuscript 'Perioperative outcomes and hospitalization costs of radical vs. conservative surgery for hepatic cystic echinococcosis: A retrospective study' has been provisionally accepted for publication in PLOS Neglected Tropical Diseases.

Best regards,

Eva Clark, M.D., Ph.D.

Section Editor

Francesca Tamarozzi

Section Editor

Reviewer's Responses to Questions

**Key Review Criteria Required for Acceptance?**

**Methods**

-Are the objectives of the study clearly articulated with a clear testable hypothesis stated?

-Is the study design appropriate to address the stated objectives?

-Is the population clearly described and appropriate for the hypothesis being tested?

-Is the sample size sufficient to ensure adequate power to address the hypothesis being tested?

-Were correct statistical analysis used to support conclusions?

-Are there concerns about ethical or regulatory requirements being met?

Reviewer #1: OK

Reviewer #3: (No Response)

**Results**

-Does the analysis presented match the analysis plan?

-Are the results clearly and completely presented?

-Are the figures (Tables, Images) of sufficient quality for clarity?

Reviewer #1: OK

Reviewer #3: (No Response)

**Conclusions**

-Are the conclusions supported by the data presented?

-Are the limitations of analysis clearly described?

-Do the authors discuss how these data can be helpful to advance our understanding of the topic under study?

-Is public health relevance addressed?

Reviewer #1: OK

Reviewer #3: (No Response)

**Editorial and Data Presentation Modifications?**

Reviewer #1: In line:

90 surgical treatments, percutaneous treatment, chemotherapy, and “watch and wait”

Please consider to replace term chemotherapy by antiparasitic (pharmacological) treatment. Chemotherapy is correct but is more likely to related to oncologic diseases.

Reviewer #3: (No Response)

**Summary and General Comments**

Reviewer #1: Dear authors. As I mentioned before this paper is a very interesting and important paper.

Radical surgery is recommended as ideal treatment and as gold standard by lower rate of recurrence but is not suitable in any place. Needs more resources as ICU and highly skilled and experienced surgeons in hepatic surgery.

The concept of conservative surgery is not only for CE, also for oncologic surgery for liver tumors, is the actual tendence. To be more conservative as possible and less aggressive. Mainly because CE is a benign pathology. So to put in balance and looking for available resources at each place, the best treatment possible in each place will be the best as possible for each particular patient and where lives.

Reviewer #3: Authors have been addressed all concerns. The quality of the manuscripts has improved considerably. Only two minor points should be addressed before it can be accepted:

1. Please include quantitative values in the abstract (results section). The current version is descriptive only.

2. In the Supplement, the cost is still based on Chinese currency. Please change it to USD, similar to the main manuscript.

PLOS authors have the option to publish the peer review history of their article (what does this mean?). If published, this will include your full peer review and any attached files.

Reviewer #1: **Yes: **Leonardo Javier Uchiumi

Reviewer #3: **Yes: **Sepehr Abbasi Dezfouli

---

## [Editor Report · Acceptance letter]

18 Oct 2024

Dear Su,

We are delighted to inform you that your manuscript, "Perioperative outcomes and hospitalization costs of radical vs. conservative surgery for hepatic cystic echinococcosis: A retrospective study," has been formally accepted for publication in PLOS Neglected Tropical Diseases.

Best regards,

Shaden Kamhawi

co-Editor-in-Chief

Paul Brindley

co-Editor-in-Chief
